# Associations of Gait Speed, Cadence, Gait Stability Ratio, and Body Balance with Falls in Older Adults

**DOI:** 10.3390/ijerph192113926

**Published:** 2022-10-26

**Authors:** Marcelo de Maio Nascimento, Élvio Rúbio Gouveia, Bruna R. Gouveia, Adilson Marques, Francisco Martins, Krzysztof Przednowek, Cíntia França, Miguel Peralta, Andreas Ihle

**Affiliations:** 1Department of Physical Education, Federal University of Vale do São Francisco, 56304-917 Petrolina, Brazil; 2Department of Physical Education and Sport, University of Madeira, 9020-105 Funchal, Portugal; 3LARSYS, Interactive Technologies Institute, 9020-105 Funchal, Portugal; 4Center for the Interdisciplinary Study of Gerontology and Vulnerability, University of Geneva, 1205 Geneva, Switzerland; 5Regional Directorate of Health, Secretary of Health of the Autonomous Region of Madeira, 9004-515 Funchal, Portugal; 6Saint Joseph of Cluny Higher School of Nursing, 9050-535 Funchal, Portugal; 7Faculty of Human Kinetics, University of Lisbon (CIPER), 1495-751 Lisbon, Portugal; 8Faculty of Medicine, University of Lisbon (ISAMB), 1649-020 Lisbon, Portugal; 9Institute of Physical Culture Sciences, Medical College, University of Rzeszów, 35-959 Rzeszów, Poland; 10Department of Psychology, University of Geneva, 1205 Geneva, Switzerland; 11Swiss National Centre of Competence in Research LIVES—Overcoming Vulnerability: Life Course Perspectives, 1015 Lausanne, Switzerland

**Keywords:** aging, vulnerability, falls, mobility, postural control, measuring instruments

## Abstract

To investigate the association between gait speed (GS), cadence (CAD), gait stability ratio (GSR), and body balance (BB) with falls in a large sample of older adults. The analysis included 619 individuals—305 men and 314 women (69.50 ± 5.62 years)—residing in the Autonomous Region of Madeira, Portugal. Mobility in GS, CAD, and GSR was assessed using the 50-foot walk test and BB by the Fullerton Advanced Balance scale. The frequency of falls was obtained by self-report. Linear regression analysis showed that higher performance in GS and BB was able to reduce the risk of falling by up to 0.34 and 0.44 times, respectively. An increase in the GSR value enhanced the risk of falling by up to 0.10 times. Multinomial analysis indicated that, in relation to the highest tertile (reference), older adults classified with GS and BB performance in the lowest tertile (lowest) had an increased chance (OR) of falling by up to 149.3% and 48.8%, respectively. Moreover, in relation to the highest tertile, the performance of the GSR classified in the lowest and medium tercile showed an increase in the chance of falling by up to 57.4% and 56.4%, respectively.

## 1. Introduction

Morbidity and mortality resulting from fall injuries in older adults is a serious public health problem worldwide [1]. It is estimated that 30–60% of older adults experience at least one fall episode per year [2]. Furthermore, up to 20% of these episodes are responsible for injuries [3], fractures, and hospitalization [4], and depending on the age group and health condition of the individual, a fall event can result in the death of the older adult [5]. Consequently, fall injuries significantly impact public and private health costs [4]. In old age, falls are also associated with mobility limitations [6], which hamper carrying out activities of daily living (ADLs) independently and safely [7]. Moreover, after a fall, because of the fear of falling again [8] and as a strategy to avoid further falls [9], older adults tend to reduce their daily movement rates, which increases the risk of developing sedentary habits [10]. For all these reasons, falls can significantly reduce the quality of life and well-being related to physical and mental health in the older adult population [8,11].

The determinants of falling are multifactorial (e.g., biological, environmental, socioeconomic, etc.). One important determinant is the performance of physical abilities such as gait speed (GS) and body balance (BB). GS, for example, is negatively correlated with age [12]. From the age of 60, the usual GS decreases by about 1% per year [13]. In older adults over 70, the prevalence of gait disorders is estimated at 35% [14]. An average GS performance is considered from 1.0 m/s, and a GS below 0.8 m/s is associated with a limited ability to walk with an increased risk of falls [15]. In turn, a performance below 0.7 m/s corresponds to an increase in the risk of falling up to 1.5 times [16]. In terms of falls, other significant parameters for ambulation tend to change with age, such as cadence, stride length, joint angular displacement, joint torque, and power [17]. Moreover, GS performance values are also useful for the early detection of cognitive impairments in older adults [18]. Thus, in the clinical area, GS can serve as a biomarker of possible changes in brain regions responsible for gait [19]. Therefore, cognitive performance and GS play an essential role in controlling body balance (BB) [20,21].

Aging also reduces postural stability [21], affecting voluntary body movement planning, including gait stability [22]. For this reason, older adults have greater difficulty perceiving external forces on the body (somatosensory function), planning body stabilization strategies (cognitive function), and therefore sending commands to the extremities of the body to create postural adjustments [23]. The systems involved in controlling BB are multifactorial, depending on the proper functioning of the visual, auditory, and somatosensory systems [24]. Furthermore, the performance of BB still depends on the efficiency of central and peripheral neurons [25], which due to physiological aging, can be affected by the reduction in skeletal mass [26], responsible for the decrease in size and number of muscle fibers mainly type II [27]. All this causes a delay in postural correction, increasing the risk of falling [28]. Comparatively, older adults are more likely than younger individuals to suffer degenerative processes in the joints and spinal and hip deformities [29]. All of this can significantly impact BB control (i.e., forward-leaning of the trunk), shifting the center of gravity [30] and consequently affecting gait biomechanics [31].

Most fall episodes among older adults occur during movement [32]. Therefore, a proper dynamic balance performance is essential to stabilize the body in its base of support [33]. To correct postural deficits during gait, older adults adapt their movement pattern, decelerating gait [34] and decreasing stride length. Thus, with each step, there is a reduction in the risk of falling due to lesser forward progression of the body [33]. Over the years, several methods have been suggested to quantify the stability of the older adult’s gait, helping to detect the risk of falling [35,36]. In general, the proposed indicators are grouped into two classes: stability and variability indices. Stability refers to the analysis of mechanical systems [37], while variability represents a manifestation of change in the gait system [35]. In this context, it is assumed that a high variability means instability for the entire system. However, despite the assumptions made for the definition of stability and variability indices, the detailed verification of both is not a simple procedure. There are also controversies about the methods applied [37]. Regarding variability, a suggested measure that considers changes in GS, also capable of reflecting changes in stride length, is the gait stability ratio (GSR) [33]. Its calculation is performed by the ratio between cadence (steps/second) and speed (meters/second), expressed in units of steps per meter. An advantage of the GSR measure is the indication of gait stability as the index increases. Thus, the more steps per unit of distance an older adult performs, the greater the proportion of their gait cycle in the phase of contact of the lower limbs with the ground. Through this strategy, older adults reduce the dynamic components of the gait, making it more stable [33].

The literature highlights that the intrinsic and extrinsic factors involved in both gait variability and stability were poorly reported [38,39]. Therefore, it is important to carry out studies in this area to qualify and expand the understanding of the underlying mechanisms responsible for falls in older adults associated with gait stability [38]. In turn, the findings can improve useful strategies for assessment, prevention, and intervention procedures, even enabling the treatment of changes in gait parameters in a way that is adjusted to the individual characteristics of each person [5]. Addressing these major gaps in the prior literature, this study aimed to investigate the association between GS, CAD, GSR, and BB with falls in a large sample of older adults. We hypothesized that (1) higher values of GS, CAD, and BB will indicate a negative association with falls, while a higher value of GSR will show a positive association with falls, (2) GS, CAD, BB, and GSR can predict falls in the population evaluated, and (3) a higher performance of GS, CAD and BB reduces the chance of falling, and the higher the value of GSR, the greater the chance of falling.

## 2. Methods

### 2.1. Design and Participants

A cross-sectional analytical observational study was conducted with 619 individuals, of which 314 were men and 305 were women (69.50 ± 5.62 years). Study members resided in different districts of the Autonomous Region of Madeira, Funchal, Portugal. The procedures for the recruitment of participants took place between January and August 2017 and were disclosed by the research team members in clubs, cultural and sports associations, and residential and public places (i.e., churches, fairs, and municipal gardens). In addition, dissemination in newspapers, radio, and television was made. Inclusion criteria were as follows: (1) residing in the community, (2) being between 60 and 79 years of age, and (3) being able to walk independently. Exclusion criteria were as follows: (1) medical contraindications for submaximal exercise, according to American College of Sports Medicine guidelines [40], and (2) inability to understand or follow investigation protocols. The evaluation procedures of the present study were performed by a team of researchers previously trained to apply the protocols. The activities took place at the Laboratory of Human Physical Growth and Motor Development at the University of Madeira. This study was scientifically and ethically approved by the Scientific Committee of the Department of Physical Education and Sports of the University of Madeira, of the Regional Secretariat for Social Affairs Committees. All participants were informed about the procedures and gave consent before taking part in the assessments. The investigation included adherence to the Declaration of Helsinki.

### 2.2. Data Collection

#### 2.2.1. Demographics and Clinical Data

Through face-to-face interviews, participants reported their sex, age, years of education, comorbidities, and the number and types of medication consumed daily.

#### 2.2.2. Falls

Through face-to-face interviews, information was collected about the history of falls in the last 12 months, with yes/no answers. If yes, the following question was asked: Can you remember how many times you fell in the last 12 months? Finally, the participants’ fear of falling was quantified. A five-digit scale (0 to 4) was used. The interpretation was as follows: zero being the absence of fear of falling and four being the maximum value of fear of falling.

#### 2.2.3. Anthropometry

Body weight and height were measured using an anthropometric scale and a Welmy^®^ stadiometer coupled to 0.1 cm and 0.1 kg [41]. Body mass index (BMI) was defined as (weight [kg])/(height [m^2^]).

#### 2.2.4. Gait

Three trials were collected for each participant. However, for the analyses, only the fastest result among the three measures was assumed. Gait parameters of interest were GS, CAD, and GSR. GS was assessed using the 50-foot walk test [42]. Specifically, participants were asked to walk a distance of 30 feet at their usual speed. The GS (m/s) calculation was obtained by dividing the 30 feet walked by the time taken to cover that distance. CAD (steps/s) was calculated by dividing the number of steps taken in space during the 30-foot walk test by the time taken to cover that distance. GSR (steps/m) was calculated by dividing the CAD value by the GS value [33]. The GSR measure quantitatively indicates the adaptation strategy used by an individual to increase gait stability. Therefore, an increase in GSR suggests an increase in the number of steps performed in one meter and a decrease in stride length. In turn, a reduction in stride length reflects a decrease in forwarding progression and a consequent increase in time spent supporting both lower limbs [43,44]. Previous studies have highlighted GS, CAD, and GSR as effective measures to assess gait adaptation strategies to maintain balance during challenges [45,46].

#### 2.2.5. Body Balance

The BB assessment was performed using the Fullerton Advanced Balance (FAB) [47] scale, which is an instrument capable of measuring the multiple dimensions of static and dynamic balance in the older adult population. Its protocol consists of 10 tasks: (1) Stand with feet together and eyes closed, (2) reach out to pick up an object (pencil) at shoulder height with the arm extended, (3) rotate 360° right and left, (4) step up and down a six-inch bench, (5) walk in tandem, (6) stand on one leg, (7) stand on foam with eyes closed, (8) long jumping, (9) walking with head turned, and (10) recovering from an unexpected loss of balance. Each test item was scored on a 4-point ordinal scale (0–4), resulting in a maximum of 40 points. The predictive validity of the FAB scale concerning the risk of falling has been presented previously [48]. A detailed description of the test administration protocol, equipment, and instructional video can be accessed [47].

### 2.3. Statistical Analysis

Initially, the data distribution was tested using the Kolmogorov–Smirnov test. After verifying the normality of all variables, scalar variables were presented by means and standard deviations (SD), while categorical variables were presented by frequency (%). Thus, differences in scalar variables between groups were processed by the parametric unpaired Student’s *t*-test. At the same time, the chi-square test (*x*^2^) and Fisher’s exact test were used to establish differences between the groups of categorical variables. The participants were divided into two groups according to the self-report on the history of falls: the faller group included participants who had between 1 to 10 fall events in the last 12 months, while the nonfaller group was formed by those without any report of falls. Data were processed in three steps to test cross-sectional associations. First, we tested whether high values of GS, CAD, and BB would show a negative association with falls and if high values of GSR would indicate a positive association with falls (first hypothesis). The correlations results were presented and interpreted by Pearson’s, using the following coefficients (*r*): 0.1 = small, 0.3 = medium, and ≥0.5 = large [49]. Second, we verified whether GS, CAD, BB, and GSR could predict falls (second hypothesis). Thus, we chose to analyze the “faller group” as a reference > Subsequently, four linear logistic regressions were run separately (insertion method): we analyzed the performance of GS, CAD, GSR, and BB (independent, continuous variable) vs. falls (dependent variable, binary).

Third, we checked in depth whether higher GS, CAD, and BB performance could reduce the chance of falling, and if the higher the GSR value, the greater the chance of falling (third hypothesis). We selected the “faller group” as a reference for this analysis. The procedures were performed separately by four multinomial logistic regressions: we included falls (dependent variable, binary) vs. the performance of GS, CAD, GSR, and BB (independent variable/tertile analysis). The use of tertiles was assumed to avoid the assumption of collinearity in the evaluation of continuous variables (motor performance). The stratification was based on the literature since it can be expected that gait parameters (GS, CAD) [34] and BB performance [21] are different between individuals because of the confounding factors that can influence the results of the analysis. The first tertile (T1) was composed of individuals with lower performance, the second tertile (T2) included those with average performance, and the third tertile (T3) was assumed as the reference category (superior performance). Only covariates with a *p*-value < 0.20 in the univariate analysis were included as control factors in the adjusted analyses. Both logistic regression analysis and multinomial regression analysis results were presented in two models: Model 1 unadjusted and Model 2 adjusted for sex, age, height, comorbidities, number of medications, and years of education. All analyses were estimated by *β*-coefficient, standard error (SE), as well as by the odds ratio (OR), accompanied by its respective confidence interval (95% CI). The statistical analyses were performed using IBM-SPSS (IBM Corp., Armonk, NY, USA), version 22.0. The significance level was defined as α < 0.05.

## 3. Results

### 3.1. Main Characteristics of the Participants

Six hundred and sixteen participants were evaluated (see Table 1 for an overview). Of these, 223 were classified as fallers, and 393 had no history of falls for the last twelve months. The mean age of the participants was 69.50 (69.50 ± 5.62) years (*p* = 0.066). Between groups, the mean age was: faller (69.18 ± 5.43) years and nonfaller (70.05 ± 5.62) years (*p* < 0.001). There was a prevalence of septuagenarians (48.19), followed by sexagenarians (47.22%) and octogenarians (3.57%). Regarding sex, 49.51% were women (*p* < 0.001). Significant differences were also found for BMI and MMSE (*p* < 0.001) but not for years of education (*p* > 0.050). Among the self-reported comorbidities, the most prevalent with levels of significance (*p* < 0.001) were visual impairment (78.90%), heart disease (65.74%), and musculoskeletal disease (7.00%). A different statistical result was not found for the BMI (*p* = 0.065). Regarding the objectively measured motor variables, comparatively, members of the FG showed better performance: GS (*p* < 0.001), CAD (*p* = 0.279), GSR (*p* < 0.001), and BB (*p* < 0.001).

### 3.2. Results of the Correlation Matrix Coefficients of the Main Study Variables

The results of the correlation matrix coefficients are presented in Table 2. A negative and small correlation was found between falls and GS (*r* = −0.174; *p* < 0.001), negative and large between GS and GSR (*r* = −0.844; *p* < 0.001), positive and large between GS and CAD (*r* = 0.789; *p* < 0.001), as well as positive and large between GS and BB (*r* = 0.561; *p* < 0.001). CAD correlated negatively and not significantly with falls (*r* = −0.044; *p* = 0.279). GSR showed a positive and small correlation with falls (*r* = 0.218; *p* < 0.001), and negative and medium with CAD (*r* = 0.412; *p* < 0.001). In turn, BB indicated a negative and small correlation with falls (*r* = −0.161; *p* < 0.001), positive and large with GS (*r* = 0.561; *p* < 0.001), positive and medium with CAD (*r* = 0.363; *p* < 0.001), and negative and large with GSR (*r* = −0.560; *p* < 0.001).

### 3.3. Associations between GS, CAD, GSR, and BB with Falls (Continuous Variable)

Table 3 presents a multiple linear regression analysis generated to assess associations between motor variables with falls. The model obtained was statistically significant [F(1,611) = 2.556; *p* < 0.001; *R*^2^ = 0.356]. The unadjusted analysis revealed a negative and significant association between GS and falls. Thus, we found that an increase in GS was able to reduce the risk of falling by up to −0.34 times (*p* < 0.001). The association between CAD and falls was negative and indicated no significant association for CAD (*p* = 0.279). GSR indicated a positive and significant association, with the risk of falling increasing to 0.44 times (*p* < 0.001). For BB, there was a negative and significant association with falls, with a fall risk reduction of up to −0.10 times (*p* < 0.001). After controlling for potential confounders (i.e., sex, age, height, comorbidities, number of medications, and years of education), the associations of most motor variables with falls were attenuated. The association of GS with falls remained negative and significant, indicating a reduction in the risk of falling by up to −0.25 times (*p* = 0.002). CAD indicated a negative and nonsignificant association with falls (*p* = 0.151). GSR showed a positive and significant association, indicating an increased risk of falling up to 0.31 times (95% CI 0.140–0.471, *p* < 0.001). Finally, BB showed a negative and significant association, indicating a reduction in the risk of falling by up to −0.07 times (*p* = 0.016).

### 3.4. Associations between GS, CAD, GSR, and BB with Falls (Terciles)

Regarding the multinomial analysis (see Table 4 for an overview), the unadjusted model indicated a positive and significant association for GS, revealing that a performance classified in T1 (lowest) increased the risk of falling by 0.91 (*p* < 0.001) times more than those classified in the upper tertiles. On the other hand, higher GS values did not present a risk for falls. There was a positive and nonsignificant association for T2 (*p* = 0.228). CAD showed a positive and nonsignificant association for T1 (*p* = 0.335), and a negative and nonsignificant association for T2 (*p* = 0.853). These results indicated that CAD was not considered a predictor of falls, at least when used alone. On the other hand, when associated with GS (GSR calculation), CAD was considered a significant predictor of falls. For GSR, participants classified at T1 (lowest) indicated a −0.85 (*p* = 0.001) times lower risk of falling than those ranked in the higher tertiles. In turn, older adults classified with GSR in T2 (medium) presented a −0.83 (*p* < 0.001) lower risk of falling than those ranked in the higher tertiles. Concerning BB, the analysis showed a positive and significant association for those classified as T1 (lowest), indicating 0.72 (*p* < 0.001) times greater risk for falls than those classified with BB in higher tertiles. A performance in T2 (medium) indicated a nonsignificant result for falls (*p* = 0.058), attesting to the protective role of high balance performance in preventing falls.

After controlling for potential confounders (i.e., sex, age, height, comorbidities, number of medications, and years of education), the adjusted model indicated a positive and significant association for older adults classified in T1 (lowest). Therefore, 0.70 (*p* = 0.002) times were more likely to develop the risk of falling, concerning being classified in the highest tertile. Showing that higher values of GS do not present a risk for falls, there was a nonsignificant association for those classified in T2 (*p* = 0.842). Confirming that the CAD measure was not a predictor of the risk of falling, the analysis showed a nonsignificant association between CAD and falls for those classified in T1 (*p* = 0.183) and those classified in T2 (*p* = 0.730). Regarding the GSR, the analysis showed a nonsignificant association for those classified as T1 (lowest) (*p* = 0.121). On the other hand, it revealed a negative and significant association for those classified in T2 (medium), indicating −0.50 (*p* = 0.027) a lower risk of falling than those classified in the higher tertiles. For a BB performance in T1 (lowest), there was a positive and significant association, showing a 0.48 (*p* = 0.036) times greater risk of falling compared with those classified in the highest tertile. On the other hand, we found that a BB performance in the medium tertile (T2) indicated a nonsignificant association for falls (*p* = 0.268).

## 4. Discussion

Our study investigated the association between falls with GS, CAD, and BB in a large sample of older adults. Our first hypothesis was confirmed. We found negative and significant associations between GS and BB with falls. A possible explanation for the small correlation coefficients may be related to the fact that falls are events with multifactorial causes [50] resulting from intrinsic and extrinsic factors [51]. Therefore, motor skills such as GS and BB can potentiate a fall event. However, it should be noted that both are part of a set of factors contributing to the increased risk of falling in this population [52,53]. Moreover, the analysis pointed to a positive and significant association between GSR and falls, which is in line with previous studies [33,46]. On the other hand, we did not find a significant result between CAD and falls. A possible explanation is that the CAD performance of both groups was adequate, not representing fall risk. This finding was highlighted by a low and nonsignificant correlation coefficient between CAD and falls (see Table 2 for an overview). In the screening procedures used to identify older adult fallers and differentiate them from nonfallers, CAD and step length are crucial measures to improve GS in rehabilitation programs for frail older adults [54,55].

We partially confirm our second hypothesis on the basis of the significant and negative results revealed by the unadjusted linear regression analysis. Thus, except for the CAD measure, all other tests proved to be fall predictors. It was found that the high performance of GS and BB were associated with lower odds of falls and, therefore, with a protective effect capable of reducing the chances of falling up to 82.6% and 83.9%, respectively. On the other hand, a high GSR value was associated with an increase in the likelihood of falling by up to 78.2%. In the geriatric population, gait is one of the most critical dynamic activities to maintain an independent daily life. Therefore, gait instability is an important fall risk factor [35]. The term stability is considered the behavior of a system under minimal perturbations [56]. In this sense, after being disturbed, a system can remain stable when it can recover a state of equilibrium in a static situation or maintain its state of uniform motion [17]. Thus, understanding gait quantified parameters can help estimate the risk of falling and plan fall prevention strategies [52]. When controlled for confounding factors (i.e., sex, age, height, comorbidities, number of medications, and years of education), high performance of GS and BB also indicated a reduction in the chances of falling by up to 87.2% and 89.8%, respectively. Following the order of the results, the CAD measure remained nonsignificant. In turn, a high GSR value indicated an increase in the chance of falling by up to 85.0%.

The GSR measurement provided a quantitative indication of the adaptation strategy performed by the evaluated population to increase gait stability [33]. Thus, our results showed that each increase in standard deviation (SD) of 0.1 m/s in GSR determined a decrease in stride length (i.e., slower forward progression of the center of mass) and, consequently, an increase in the percentage of double limbs stance time in the gait cycle [46]. So, GS and CAD tend to decrease. In the present study, it is likely that the increase in GSR was a reflection of the reduction in GS and not in CAD performance because it presented nonsignificant results in all analyses. Moreover, the high GSR value was more conclusive for the older adult with a history of falls (see Table 1 for an overview). A previous study showed that after a gait disturbance, faster recovery from older adult BB was associated with better lower-limb coordination rather than better gait stability [44]. It is suggested that the ability of older adults to coordinate limb stability according to gait has distinct locomotor characteristics. Therefore, when it comes to falls, both variables must be considered in assessing the risk, mainly because falls are multifactorial events.

Another point to consider is that the GSR showed a strong and negative correlation with the BB, suggesting that the higher the GSR, the lower the older adult’s ability to stabilize their posture during gait. For this reason, fallers took shorter steps, walked more slowly, and presented a more significant deficit in dynamic balance, which consequently compromised the dynamic stability of gait (higher GSRs). In general, to improve dynamic gait stability, older adults assume a more favorable posture, shifting the center of mass forward [57], which is accompanied by an increase in the velocity of the center of mass affecting the support base [12]. Thus, during gait, both the positioning of the feet on the ground and their alterations play a substantial role in stabilizing the BB [12]. In the present study, we found changes in GS, BB, and GSR for the entire sample evaluated. Lee et al. [12] evaluated 184 community-dwelling older adults (65 years over), founding a significant decline in GS only for those aged >85 years compared with those aged 65–69 years, attributing the findings to a likely reduction in step length and CAD performance. The authors also found a decrease in GSR and a greater risk of falling backward after 85 years because of a reduction in the velocity of the center of mass to the base support. Moreover, when it comes to BB stabilization, it should be considered that the ankle muscles’ mechanics are essential to upright posture [58]. Therefore, muscle weakness in this region can potentiate postural instability, increasing the risk of falling [59].

According to the unadjusted multinomial analysis, our third study hypothesis was also partially confirmed. With the exception of CAD, the results confirmed that the high performance of GS and BB represented a lower chance of falling. Therefore, improving these motor skills plays a protective role in preventing falls. Thus, for the older adult with performance values classified as T1 (lowest), there was an increase in the chance of falling by up to 149.3%. In relation to BB, participants classified in T1 (lowest) indicated an increase in the chance of falling by up to 48.8%. In turn, in relation to the upper tertile, the GSR showed an increase in the chance of falling for older adults classified in T1 (lowest) of up to 57.4% and up to 56.4% for those classified in T2 (medium). Thus, we concluded that, regardless of the tertile, the GSR was a predictor of decline. Furthermore, the lower the tertile rank, the lower the chance of an older adult falling. The findings confirmed the strategy adopted by the participants to stabilize gait in an attempt to compensate for the BB deficit [33,46]. Thus, they performed more steps per unit of distance, reaching a gait pattern more resistant to external disturbances—a strategy adopted to avoid falls.

When controlled for confounding factors (i.e., sex, age, height, comorbidities, number of medications, and years of education), it was observed that the performance of the GS classified in T1 (lowest) indicated an increase in the chance of falling by up to 101.6%. In the case of BB, confirming our assumptions, a performance in T1 (smaller) showed an increase in the chance of falling by up to 61.4%. Regarding the performance of GS and BB in T2 (medium), we did not find significant results. The findings showed that the high performance of GS and BB were not predictors of falls. Attesting that a GSR performance in T1 (lower) was not associated with the fall, we did not find a significant result. On the other hand, older adults classified as T2 (medium) indicated a maximization of gait stability and a consequent increase in the chance of falling by up to 39.2%. These findings confirmed that quantitative gait markers are independent predictors of falls in the older adult population [16], capable of contributing to the procedures for detecting falls [60] and qualifying intervention strategies [52].

Some limitations should be noted. First, although we analyzed a large and representative sample of older adults, because of the cross-sectional design, caution is suggested in generalizing the results with regard to changes over time. Second, it is known that among older adults, gait stability is associated with physical activity levels [61], including lower limb strength [38,62]. Thus, considering that our analysis did not control the participants’ level of physical activity and that the sample was recruited from different locations in the city, older adults may have unequal physical conditions. Therefore, this may have resulted in further interindividual differences in the performance of GS, CAD, BB, and, consequently, the GSR values. Third, although the fear of falling (FOF) did not indicate a significant difference in the analysis between fallers and nonfallers, it is known that FOF is strongly associated with the risk of falling [63]. Moreover, it is worth noting that, in the present study, the evaluation of FOF was carried out through a five-digit scale (0-4) and not with a validated and specific questionnaire for FOF. Thus, it is suggested that future studies evaluate the FOF and that the procedures take place using instruments specific to the examination of the FOF. Fourth, it is known that specific age and sex differences should be considered when measuring objective gait parameters [64]. Thus, a possible focus for future investigations may be the association between the adaptation of gait stabilization (GSR) related to different age groups, deepening the analysis according to sex. Finally, another essential point considered in the context of older adult mobility is cognitive functioning and perhaps also cognitive reserve [64,65].

## 5. Conclusions

The findings revealed in this large sample of older adults residing in the Autonomous Region of Madeira, Portugal, highlighted their strategies to adapt, compensate, and correct deficits in BB performance during gait, which suggests a strategy to prevent accidents and falls. In line with previous studies, our quantitative analysis evidenced the protective role that the high performance of GS and BB can offer against falls. Unlike previous studies, the CAD measure did not prove to be a predictor of falls when used alone. On the other hand, we confirm previous studies that an increase in the GSR value was associated with an attempt to stabilize the gait. Thus, comparatively, older adults with a history of falls had higher GSR values than those without a history of falls. Our findings may provide important information about the mechanisms older adults use to adapt to gait stability because of the increased imbalance in postural control. Thus, our information can serve as a basis for comparative analyses of future investigations. Moreover, the findings can help clinicians to plan gait and balance training programs capable of preventing or minimizing the incidence of falls and sequelae in older adults.

## Figures and Tables

**Table 1 ijerph-19-13926-t001:** Main characteristics of the sample.

Variable	Full Sample (*n* = 619)	Faller (*n* = 225)	Nonfaller (*n* = 394)	*p*-Value
Age (years)	69.50 ± 5.62	70.05 ± 5.62	69.18 ± 5.43	0.066
Age group n (%)				
60–69 years	294 (47.50)	104 (46.22)	190 (48.22)	
70–79 years	303 (48.95)	110 (48.88)	193 (48.98)	
80–89 years	22 (3.55)	11 (4.88)	11 (2.79)	
Sex n (%)				<0.001
Women	305 (49.27)	153 (68.00)	161 (40.86)	
Men	314 (50.72)	72 (32.00)	233 (59.13)	
Medication (n)	4.65 ± 0.97	4.61 ± 0.94	4.68 ± 1.00	0.395
Education (years)	6.75 ± 4.52	6.56 ± 4.20	6.94 ± 5.82	0.128
Comorbidities n (%)				
Vision	486 (78.51)	204 (90.66)	282 (71.57)	<0.001
Hearing	196 (31.66)	76 (33.77)	120 (30.45)	0.228
Hypertension	405 (65.43)	152 (67.55)	253 (64.21)	0.018
Diabetes	221 (35.70)	102 (45.33)	119 (30.20)	0.131
Musculoarticular	43 (6.95)	21 (9.33)	22 (5.58)	0.035
Fear of falling (n)	2.27 ± 0.53	2.31 ± 0.62	2.24 ± 0.44	0.121
BMI (kg/m^2^)	29.56 ± 4.38	29.98 ± 4.52	29.31 ± 4.29	0.065
Gait speed (n)	1.25 ± 0.24	1.20 ± 0.25	1.28 ± 0.24	<0.001
Cadence (n)	1.92 ± 0.221	1.90 ± 0.22	1.92 ± 0.21	0.279
Gait stability ratio (n)	1.56 ± 0.23	1.63 ± 0.26	1.52 ± 0.21	<0.001
Body balance (n)	29.55 ± 4.38	30.92 ± 7.15	31.79 ± 6.72	<0.001

Kg—kilogram; cm—centimeter; BMI—Body Mass Index; Kg/m^2^—kilogram divided by meter squared.

**Table 2 ijerph-19-13926-t002:** Correlation matrix coefficients between the studied variables.

Variable	1	2	3	4
1. Falls	1.00			
2. Gait speed	−0.174 *	1.00		
3. Cadence	−0.044 ^ns^	0.789 *	1.00	
4. Gait stability ratio	0.218 *	−0.844 *	−0.412 *	1.00
5. Body balance	−0.161 *	0.561 *	0.363 *	−0.560 *

^ns^—not significant; * *p* < 0.001.

**Table 3 ijerph-19-13926-t003:** Associations between gait speed, cadence, gait stability ratio, body balance *versus* falls, results expressed by logistic regressions.

Variable	*β* (SE)	Unadjusted OR (95% CI)	*p*-Value	*β* (SE)	Adjusted OR (95% CI)	*p*-Value
GS (m/s)	−0.34 (0.077)	−0.174 (−0.491–0.187)	<0.001	−0.25 (0.080)	−0.128 (−0.408–0.092)	0.002
CAD (m/s)	−0.10 (0.090)	−0.044 (−0273–0.079)	0.279	−0.13 (0.091)	−0.059 (−0.308–0.048)	0.151
GSR (m/s)	0.44 (0.080)	0.218 (0.287–0.601)	<0.001	0.31 (0.084)	0.150 (0.140–0.471)	<0.001
BB (n)	−0.10 (0.034)	−0.161 (−0.016–0.006)	<0.001	−0.07 (0.018)	−0.102 (−0.018–0.001)	0.016

GS—gait speed; CAD—cadence; GSR—gait stability ratio; BB—body balance; OR —Odds ratio; SE—Standard error; m—meter; s—second; Linear regression, model adjusted for sex, age, height, comorbidities, and years of education, comorbidities. *p* < 0.050.

**Table 4 ijerph-19-13926-t004:** Associations between gait speed, cadence, gait stability ratio, body balance versus falls, results expressed by multinomial logistic regressions.

Variable	*β* (SE)	Unadjusted OR (95% CI)	*p*-Value	*β* (SE)	Adjusted OR (95% CI)	*p*-Value
Gait speed (m/s)						
Tertile 3 (highest)		1			1	
Tertile 2 (medium)	0.25 (0.209)	1.286 (0.854–1.937)	0.228	0.04 (0.222)	1.045 (0.676–1.616)	0.842
Tertile 1 (lowest)	0.91 (0.207)	2.493 (1.662–3.739)	<0.001	0.70 (0.226)	2.016 (1.294–3.142)	0.002
Cadence (m/s)						
Tertile 3 (highest)		1			1	
Tertile 2 (medium)	−0.03 (0.201)	1.226 (0.650–1.429)	0.335	0.07 (0.210)	1.075 (0.712–1.624)	0.730
Tertile 1 (lowest)	0.20 (0.211)	0.963 (0.811–1.853)	0.853	0.30 (0.229)	1.356 (0.866–2.123)	0.183
Gait stability ratio (m/s)						
Tertile 3 (highest)		1			1	
Tertile 2 (medium)	−0.83 (0.267)	0.436 (0.252–0.718)	<0.001	−0.50 (0.226)	0.608 (0.884–2.062)	0.027
Tertile 1 (lowest)	−0.85 (0.213)	0.426 (0.287–0.661)	0.001	−0.44 (0.284)	0.644 (1.300–3.330)	0.121
Body balance (n)						
Tertile 3 (highest)		1			1	
Tertile 2 (medium)	0.72 (0.204)	2.065 (0.996–2.223)	0.058	0.24 (0.216)	1.270 (0.832–1.939)	0.268
Tertile 1 (lowest)	0.40 (0.205)	1.488 (1.383–3.082)	<0.001	0.48 (0.228)	1.614 (1.033–2.523)	0.036

OR—Odds ratio; SE—Standard error; m—meter; s—second; Gait speed = T1 < 1.09, T2 = 1.10–1.26, T3 ≥ 1.27; Cadence = T1 < 1.79, T2 = 1.80–1.93, T3 ≥ 1.94, Gait stability ratio = T1 < 1.37, T2 = 1.38–1.50, T3 ≥ 1.51, Body balance = T1 < 27.00, T2 = 28.00–32.00, T3 ≥ 33.00. Multinomial regression model adjusted for sex, age, height, comorbidities, years of education, and comorbidities. *p* < 0.050.

## Data Availability

The data presented in this study are available upon request from the corresponding author.

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
