# Peer review of "Associations of Gait Speed, Cadence, Gait Stability Ratio, and Body Balance with Falls in Older Adults"

_ijerph, 2022, doi:10.3390/ijerph192113926_

Round 1
Reviewer 1 Report
This research conducted the analysis included 619 individuals, 305 men and 314 women (69.50 ± 5.62 years), residing in the Autonomous Region of Madeira, Portugal. Mobility in GS, CAD, and GSR was assessed using the 50-foot Walk test and BB by Fullerton Advance Balance scale. The frequency of falls was obtained by self-report. Linear regression analysis indicated for GS and BB, respectively, a reduction in the risk of falling by up to 0.34 and 0.44 times, while GSR indicated an increase by up to 0.10 times. Multinomial analysis indicated that GS and BB performance in the lowest tertile represented a chance (OR) of falling by up to 149.3% and 48.8%, respectively. GSR performance classified in the lowest and medium tertile indicated an increase in the chance of falling by up to 57.4% and 56.4%, respectively. Finaly, poor GS and BB performance was associated with a higher risk of falling, CAD was not relevant for falls, and GSR proved to be a useful measure to identify adaptations in gait variability due to BB control deficit. Good work overall.
Author Response
Dear Reviewer, thank you for your comments!
Reviewer 2 Report
Article of growing interest due to its subject matter. Falls in older adults are an important topic of study due to their high incidence and comorbidity. Knowing more about them will help us to better understand what preventive mechanisms we can implement.
As recommendations of the study I propose:
1. Improve the methodological procedure to be followed; it is not entirely clear where the interventions are carried out, nor the time frame of the interventions.
2. In the discussion, hypotheses are mentioned but they are not correctly described in the manuscript.
3. The presentation of the objectives of the study should be improved.
I suggest that with minor corrections the manuscript could be considered for publication.
Author Response
As recommendations of the study I propose:
1. Improve the methodological procedure to be followed; it is not entirely clear where the interventions are carried out, nor the time frame of the interventions.
Reply
Dear Reviewer, in the "2.1. Design and participants" section, we have included information about the period of data collection, as well as the location, where it took place.
2. In the discussion, hypotheses are mentioned but they are not correctly described in the manuscript.
Reply
Dear Reviewer, we thank you for your observation, we revised the hypotheses, and in the Discussion section the interpretation of the results was also corrected and adjusted according to the formulated hypotheses; in particular, the interpretation of OR proportions.
3. The presentation of the objectives of the study should be improved.
Reply
Dear Reviewer, we thank you for your observation, the objectives have been revised both in the Abstract and at the end of the Introduction.
I suggest that with minor corrections the manuscript could be considered for publication

Reviewer 3 Report
This study primarily examines how gait speed, cadence, gait stability ratio and body balance influence the risk of falling in relatively large samples of male and female vulnerable older adults. The findings are interpreted and discussed within the context of relevant literature, and logical conclusions are drawn. Limitations of such study are also considered, though the authors could have also addressed the limitations of this methodological approach. The key findings look promising and have important implications, and this paper would be an important addition to this body of literature.
Specific comments:
Abstract – Lines 32-33 – Could the author please report accurately this key finding “Linear regression analysis indicated for GS and BB, respectively, a reduction in the risk of falling by up to 0.34 and 0.44 times, while GSR indicated an increase by up to 0.10 times”.
Line 48: How is vulnerable older adults defined?
Line 157: Clarify what the authors mean by “best performance was assumed” and the criteria to reject the other two trials.
Line 178: Typo – “(7) stand on foam …”
Line 193: It would be helpful to know the time of occurrence of the falls over the past 12 months. For example, a fall 12 months ago may not have the same implication as a fall a month or a few weeks ago – there is here the element of fear of falling that needs to be considered.
Author Response
1. Abstract – Lines 32-33 – Could the author please report accurately this key finding “Linear regression analysis indicated for GS and BB, respectively, a reduction in the risk of falling by up to 0.34 and 0.44 times, while GSR indicated an increase by up to 0.10 times”.
Reply
Dear Reviewer, in the Abstract section, we rewrite the findings revealed by linear regression analysis. Thank you for your observation!
2. Line 48: How is vulnerable older adults defined?
Reply
Dear Reviewer, thank you for your question, which made us reflect on keeping the word vulnerability throughout the text, including the title of the article. We don't really assess "vulnerability". Therefore, this term was removed from the manuscript, and we work exclusively with the term "old adult"
3. Line 157: Clarify what the authors mean by “best performance was assumed” and the criteria to reject the other two trials.
Reply
Dear Reviewer, in lines 169-170 this term has been clarified as well as the sentence rewritten.
4. Line 178: Typo – “(7) stand on foam …”
Reply
Dear Reviewer, on line 191 the word has been corrected. Thankful for your attention!
5. Line 193: It would be helpful to know the time of occurrence of the falls over the past 12 months. For example, a fall 12 months ago may not have the same implication as a fall a month or a few weeks ago – there is here the element of fear of falling that needs to be considered
Reply
Dear Reviewer, we keep in lines 205-208 the information on how the groups were composed, in relation to the history of falls. However, we have created our own section for "falls" (section 2.2.2.). Here, we also present the results of a face-to-face questionnaire question related to fear of falling (FoF): scale 0-4 digits.
- We have included the result of this question in Table 1. As the analysis did not indicate a significant difference, we did not explore this issue in the Results section;
- However, considering the importance of FoF in the context of older adult falls, we have included/detailed the facts in the limitation section;
2.1 It is worth noting that in the present study, the FoF was not investigated using a specific questionnaire for the case. So we also highlight this in the limitations section.
